

# White matter hyperintensities and the risk of vascular dementia: a systematic review and meta-analysis

Wei Luo[1,2], Zhiqiang Dai[3], Wenjing Wu[1,2], Haitao Li[1,2] and Yang Zhang[1,2]

[1] Department of Radiology, Southwest Hospital, Army Medical University (Third Military Medical University), Chongqing, China
[2] 7T Magnetic Resonance Imaging Translational Medical Center, Southwest Hospital, Army Medical University (Third Military Medical University), Chongqing, China
[3] Department of Neurosurgery, 920th Hospital of Joint Logistics Support Force of the Chinese People's Liberation Army, Kunming, China

## ABSTRACT

**Background.** White matter hyperintensities (WMHs) are hyperintense lesions observed on magnetic resonance imaging (MRI) and are unique imaging indicators of cerebral small vessel diseases. WMH-related white matter alterations have been correlated with cognitive impairment and cerebrovascular pathology. Some studies suggest that vascular hemodynamic changes contribute to WMH development, ultimately leading to vascular dementia (VaD). However, the association between WMH burden and VaD remains inconclusive. This meta-analysis aimed to quantify the relationship between WMH volume and VaD severity and to clarify the role of WMHs in VaD pathogenesis.

**Methods.** A systematic literature search was performed using the MEDLINE, EMBASE, and Cochrane Library databases. A total of 15 studies with 4,061 patients were selected. The meta-analysis was performed using the RevMan software (version 5.4) and Stata software (version 14.0). All the patients underwent brain MRI to assess WMH volumes or levels, and compared the differences in WMH levels among the VaD group, the non-cognitively impaired (NCI) group, the cognitively impaired no dementia (CIND) group, and the Alzheimer's disease (AD) group.

**Results.** The meta-analysis showed that all patients in the VaD group had high white matter signals on brain MRI. They also had higher WMH volumes compared to patients in the NCI, CIND, and AD groups. WMH correlated with cerebrovascular pathology, with irregular and periventricular WMHs being more specific to VaD. Sensitivity analyses were performed to identify sources of heterogeneity, while funnel plot and Egger's test suggested potential publication bias.

**Conclusions.** Patients with VaD exhibit significantly greater WMH than those with AD, NCI, and CIND, reinforcing the role of cerebrovascular pathology in VaD. These findings emphasize the need for standardized imaging assessments, multi-modal biomarkers, and the development of predictive models to enhance early diagnosis, personalized risk assessment, and targeted therapeutic strategies for VaD.

Corresponding author
Yang Zhang, yy_9981@sohu.com

## INTRODUCTION

Vascular dementia (VaD) is the second most common type of dementia after Alzheimer's disease (AD), accounting for approximately 15–20% of dementia cases worldwide (*Kalaria, 2018*). It is primarily caused by cerebrovascular pathology, including ischemic strokes, small vessel disease, and white matter hyperintensities (WMHs), all of which contribute to progressive cognitive decline (*Wahlund et al., 1994*). Among these, cerebral small vessel disease (CSVD), is a major contributor to VaD (*Wardlaw et al., 2017*). CSVD has been suggested to be associated with cognitive deficits, depression, stroke, and other neurological diseases in the aging population (*Hainsworth, Markus & Schneider, 2024*; *Sperber et al., 2024*). The pathogenesis of VaD is closely associated with chronic cerebral hypoperfusion, blood–brain barrier dysfunction, and impaired neurovascular coupling, ultimately leading to neuronal injury and cognitive impairment (*Gorelick et al., 2011*).

Vascular endothelial damage occurs from various causes, including hypertension, chronic infection, inflammation and a high-salt diet (*Conijn et al., 2011*; *Ihara & Yamamoto, 2016*; *Wardlaw, Smith & Dichgans, 2019*). This damage leads to the destruction of the blood–brain barrier (BBB), which is the key pathophysiological mechanism for the occurrence of CSVD (*Bir & Kelley, 2022*; *Wardlaw, Smith & Dichgans, 2013*). CSVD is detectable by magnetic resonance imaging (MRI), and white matter hyperintensity (WMH) is a unique feature of CSVD (*Debette & Markus, 2010*; *Shahid et al., 2024*). Other abnormalities such as subcortical infarcts, brain atrophy, cerebral microbleeds and inflamed perivascular spaces are observed in CSVD patients (*Kalaria, 2012*; *Karch et al., 2013*).

Numerous studies have investigated MRI T2-weighted WMHs as markers of small vessel cerebrovascular pathology and ischemia. The prevalence and severity of WMHs are associated with cardiovascular risk factors, aging and cognitive impairment, particularly in individuals with mild cognitive impairment, VaD and other cerebrovascular diseases. WMHs particularly affect executive function, with additional impacts on memory and global cognition (*Altamura et al., 2016*; *Habes et al., 2016*; *Hase et al., 2018*; *Silbert et al., 2008*; *Sperber et al., 2024*).

WMHs are widely recognized as markers of CSVD and are frequently observed in MRI scans. Research has suggested that it is associated with the incidence of cognitive deficits (*Kloppenborg et al., 2014*; *Sivakumar et al., 2017*). As WMHs have various clinical symptoms and imaging displays, it is difficult to detect WMHs at the early stage. Furthermore, the mechanism underlying the development of WMH remains unclear, making it difficult to prevent and treat WMH-related diseases. The probable pathological mechanism is nerve fiber demyelination due to chronic cerebral ischemia caused by CSVD (*Pendlebury & Rothwell, 2019*). WMHs have been suggested to be related to the incidence and pathological mechanisms of VaD. WMH-related cognitive decline is mainly characterized by slowed information processing, executive dysfunction, and ultimately progression to dementia (*Brainin et al., 2015*; *Gunning-Dixon & Raz, 2000*).

Although the relationship between WMHs and VaD has been extensively studied, the available evidence remains inconclusive (*Hu et al., 2021*). Some studies have reported an association between WMH burden and VaD, suggesting that vascular hemodynamic

changes contribute to white matter alterations, ultimately, leading to VaD or cognitive deficits (*Ye et al., 2018*). However, other studies have found no significant relationship. This inconsistency may result from differences in experimental design, sample sizes, or measurement techniques among studies. To reach an objective conclusion, it is essential to perform a pooled analysis investigating the relationship between WMH and VaD *via* a comprehensive literature search.

This study aims to quantitatively assess the effect of WMH on the risk of VaD, especially the relationship between WMH volume and VaD. We conducted a meta-analysis to evaluate the association between WMH volume and VaD, severity and to clarify the significance of WMH for VaD pathogenesis. At the same time, we compared the differences in WMH volumes among VaD patients, AD patients, non-cognitively impaired (NCI) patients and cognitively impaired no dementia (CIND) patients.

## MATERIALS AND METHODS

This meta-analysis was conducted following the guidelines outlined in the Cochrane Handbook of Systematic Reviews of Interventions and the Preferred Reporting Items for Systematic Reviews and Meta-analysis guidelines (PRISMA) (*Higgins et al., 2021*; *Page et al., 2021*). The study protocol has been registered in PROSPERO under the registration code CRD42024588919.

### Study selection

A systematic literature search was performed across the EMBASE, MEDLINE, and Cochrane Library (Cochrane Database of Systematic Reviews) databases. Publications from the inception of these databases until April 9, 2024 were reviewed. The search terms included "white matter hyperintensities", "WMH", "white matter high signal", "Magnetic Resonance Imaging", "MRI", "leukoaraiosis", "Dementia, Vascular", "vascular dementia," and "VaD", both in full and truncated versions.

Following the initial search, studies investigating the relationship between WMHs and VaD were selected. The full texts of relevant studies were viewed, and inclusion was determined based on the following criteria: (1) a prospective or prospective cross-sectional and longitudinal studies with a population-based or case-control design; (2) MRI-based brain imaging for the identification of WMHs; (3) diagnosis of VaD based on established clinical criteria (*Cohen et al., 2002*); (4) explicit investigation of the relationship between WMHs and VaD; and (5) setting VaD as the dependent variable and WMH as the independent variable. Studies were excluded if they (1) examined broad neurological disorders, including Alzheimer's disease and Parkinson's disease, (2) did not examine the relationship between WMHs and VaD or (3) were categorized as short reports, literature reviews, letters, editorial commentaries, or conference abstracts.

### Data extraction and quality assessment

Two independent reviewers assessed the final set of selected articles using a standardized data extraction form. When necessary, the reviewers contacted the authors of a certain study to identify the experimental procedures. The search strategy and data extraction were

 

independently conducted by Wei Luo and Zhiqiang Dai. Any disagreements that arose during the search and selection process were discussed between Wei Luo and Zhiqiang Dai. Final decisions were made by the third-party corresponding author, Dr. Yang Zhang, who served as the referee. The following data were collected: name of the first author, publication year, country/region, age, sex ratio, research type, the detection method of VaD, and the WMH values of the VaD group and other control groups.

## Statistical analysis

Statistical analyses were performed using RevMan software (version 5.4, Cochrane Collaboration, Oxford, UK) and Stata software (version 14.0, Stata Corporation, College Station, TX, USA). A random effect model was applied for analysis. For continuous variables, the standardized mean difference (SMD) with a 95% confidence interval (CI) was calculated, and results were visualized in forest plots. If more than 10 studies were included, publication bias was assessed using a funnel plot and Egger's test (*Egger et al., 1997*). If significant publication bias was detected, trim-and-fill analysis was employed to adjust for the bias. Heterogeneity among the selected studies was assessed using the $I^2$ test, and corresponding 95% CIs were reported. Statistical significance was set at $P < 0.05$.

## RESULTS

### Characteristics of the included studies

The PRISMA flowchart is presented in Fig. 1. Our initial literature search identified 551 studies from the EMBASE, MEDLINE, and Cochrane Library databases, of which 159 were duplicates. Using relevant keywords, our primary literature search identified 392 records from the databases. A total of 361 studies were excluded from our analysis because of failure to meet the study criteria. According to the inclusion and exclusion criteria, 15 studies were finally included in the analysis. These studies published between 1994 and 2024, involved a total of 4,061 patients, with 1,243 in the treatment group (VaD), and 2,818 in the control group (*Altamura et al., 2016*; *Barber et al., 1999*; *Bokde et al., 2002*; *Chia et al., 2024*; *Gootjes et al., 2004*; *Han et al., 2021*; *Ji et al., 2017*; *Logue et al., 2011*; *Purandare et al., 2008*; *Robert et al., 2022*; *Song et al., 2022*; *Staekenborg et al., 2010*; *Tanabe et al., 1999*; *Varma et al., 2002*; *Wahlund et al., 1994*). Among these, 12 studies specifically reported on the relationship between VaD and WMH volume (Table 1). The diagnostic methods for VaD varied across the 15 studies, including NINDS-AIREN and DSM-IV.

### Meta-analysis and heterogeneity analysis

Eight studies compared WMH volume between VaD patients and those in the NCI group (*Chia et al., 2024*; *Gootjes et al., 2004*; *Han et al., 2021*; *Ji et al., 2017*; *Logue et al., 2011*; *Robert et al., 2022*; *Song et al., 2022*; *Tanabe et al., 1999*). A random effects model and SEM analysis demonstrated that the WMH volume was significantly higher in the VaD group than in the NCI group (SMD = 1.33; 95% CI [1.03–1.62], $P = 0.0009$, $I^2 = 72\%$; Fig. 2A). Sensitivity analysis identified the study by *Logue et al. (2011)* as the primary source of heterogeneity. When this study was excluded, heterogeneity was reduced to 41%, and the effect size changed to SMD = 1.41, 95% CI [1.14–1.69], $P = 0.12$, $I^2 = 41\%$ (Fig. 2B). The

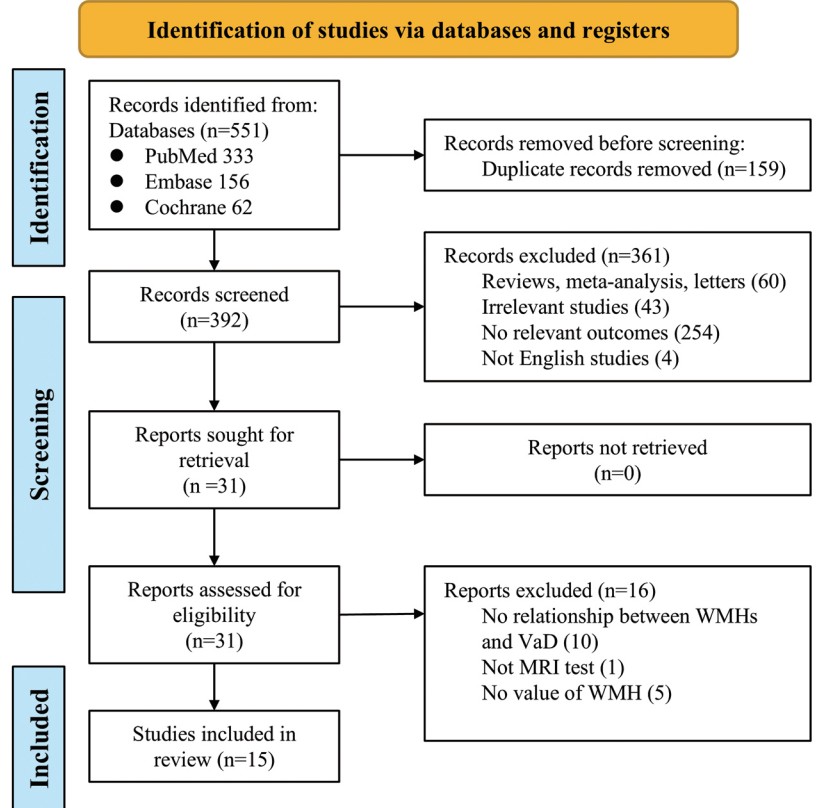

**Figure 1** **Prisma flowchart of the meta-analysis.**

high heterogeneity may be attributed to the large sample size in the study by *Logue et al. (2011)*.

Four studies examined differences in WMH volume between VaD and CIND patients (*Chia et al., 2024*; *Robert et al., 2022*; *Song et al., 2022*; *Staekenborg et al., 2010*). Meta-analysis using a random effects model and SEM analysis, indicated that WMH volume was significantly higher in VaD patients compared to CIND patients (SMD = 1.34, 95% CI [0.78–1.91], $P = 0.002$, $I^2 = 80\%$; Fig. 3A). Sensitivity analysis identified (*Staekenborg et al., 2010*) as the main contributor to heterogeneity. After excluding this study, heterogeneity reduced to 31%, and the effect size was reduced to SMD = 1.07, 95% CI [0.72–1.43], $P = 0.24$, $I^2 = 31\%$ (Fig. 3B). The high WMH volume reported by *Staekenborg et al. (2010)* may have contributed to the observed heterogeneity.

Eleven studies compared WMH volume between VaD and AD patients (*Altamura et al., 2016*; *Chia et al., 2024*; *Gootjes et al., 2004*; *Han et al., 2021*; *Ji et al., 2017*; *Logue et al., 2011*; *Purandare et al., 2008*; *Robert et al., 2022*; *Song et al., 2022*; *Staekenborg et al., 2010*; *Wahlund et al., 1994*). The random effects model and SEM analysis showed that WMH volume was significantly higher in VaD patients than in AD patients (SMD = 0.93, 95% CI [0.61–1.25], $P < 0.00001$, $I^2 = 88\%$; Fig. 4A). Sensitivity analysis identified (*Logue et al., 2011*; *Staekenborg et al., 2010*) as the largest sources of heterogeneity. Excluding these

**Table 1  Characteristics of the included studies.** Note. *Robert et al. (2022), Altamura et al. (2016), Ji et al. (2017), Han et al. (2021), Tanabe et al. (1999), Gootjes et al. (2004), Logue et al. (2011), Purandare et al. (2008), Chia et al. (2024), Staekenborg et al. (2010), Wahlund et al. (1994), Song et al. (2022), Barber et al. (1999), Bokde et al. (2002), Varma et al. (2002).*

| First author | Year | Country | Research type | VaD diagnosed by | Cases (n) | Sex (% female) | Age (mean ± SD) | Total WMH volume (mean ± SD) (mL) |
|---|---|---|---|---|---|---|---|---|
| Caroline Robert | 2022 | Singapore | Retrospective | NINDS-AIREN | 21 | 33.3% | 74.67 ± 7.07 | 10.29 ± 3.49 |
| Claudia Altamura | 2016 | Italy | cross-sectional study | NINDS-AIREN | 31 | 29.0% | 80.7 ± 3.8 | 146.07 ± 355.8 |
| Fang Ji | 2017 | Singapore | Retrospective | NINDS-AIREN | 19 | 57.9% | 74.1 ± 7.2 | 9.2 ± 4.6 |
| Ji Won Han | 2021 | South Korea | Retrospective | DSM-IV | 82 | 67.1% | 77.37 ± 5.31 | 47.98 ± 31.55 |
| Jody L. Tanabe | 1999 | US | Retrospective | MRI, clinical dementia rating | 15 | 40.0% | 79.5 ± 5.6 | 2.40 ± 1.40 |
| L. Gootjesa | 2004 | Germany, Netherlands | Retrospective | NINDS-AIREN | 20 | 35.0% | 72.3 ± 10.2 | 24.53 ± 28.19 |
| M. W. Loguea | 2011 | US | Retrospective | NINDS-AIREN | 826 | 60.2% | 73.52 ± 9.14 | 30.32 ± 26.37 |
| N Purandare | 2008 | UK | Retrospective | NINDS-AIREN | 51 | 47.1% | 76.9 ± 6.9 | 9.3 ± 6.4 |
| Rachel S. L. Chia | 2024 | Singapore | Case–control, cross-sectional study | NINDS-AIREN | 23 | 34.8% | 75 ± 9 | 14.6 ± 21.6 |
| S.S. Staekenborg | 2010 | Netherlands | Retrospective | NINDS-AIREN | 34 | 32.4% | 71 ± 9 | 45.6 ± 31.4 |
| Wahlund, L. O. | 1994 | Sweden | Retrospective | DSM-III-R diagnostic criteria, NINCDS-ADRDA | 31 | 71.0% | 79 ± 1 | 1.89 ± 1.7 |
| Yang Song | 2022 | China | Retrospective | NINDS-AIREN | 24 | 25.0% | 65.96 ± 7.22 | 4.54 ± 1.53 |
| R Barber | 1999 | UK | Retrospective | NINCDS/ADRDA and dementia with Lewy bodies consensus criteria | 25 | 40.0% | 76.8 ± 7 | WMH frontal = 5, 5≥11 mm |
| A.L.W. Bokde | 2002 | Germany | Case-control study | NINDS-AIREN | 10 | | 67.7 ± 7.8 | WMH densities 10-4/mm: Frontal: 44.83 ± 61.26; Temporal: 3.69 ± 2.73; Parietal: 50.91.23 ± 55.58; Occipital: 11.62 ± 12.46 |
| Varma AR | 2002 | UK | Retrospective | NINCDS-ADRDA | 31 | 45.2% | 65.41 ± 7.96 | Grade 1: 6 ± 19.4; Grade 2: 12 ± 38.7; Grade 3: 5 ± 16.1 |

two studies reduced heterogeneity to 49% and resulted in an adjusted effect size of SMD = 0.85, 95% CI [0.62–1.08], $P = 0.05$, $I^2 = 49\%$ (Fig. 4B). The high sample sizes in these studies may explain the observed heterogeneity.

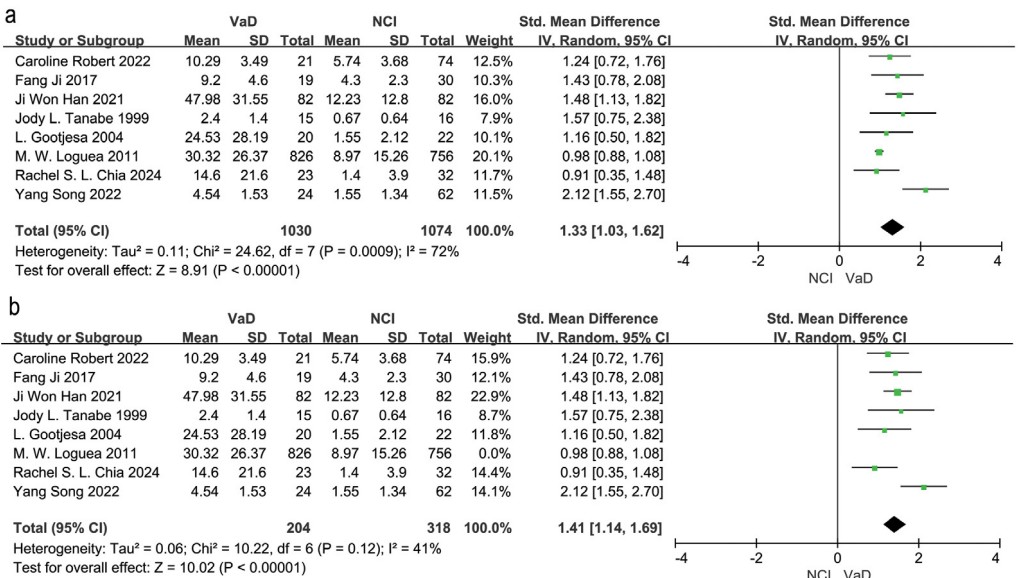

**Figure 2  Meta-analyses forest plot of WMH volume between patients with VaD and NCI.** Note. *Robert et al. (2022)*, *Ji et al. (2017)*, *Han et al. (2021)*, *Tanabe et al. (1999)*, *Gootjes et al. (2004)*, *Chia et al. (2024)*, *Song et al. (2022)*, *Logue et al. (2011)*.

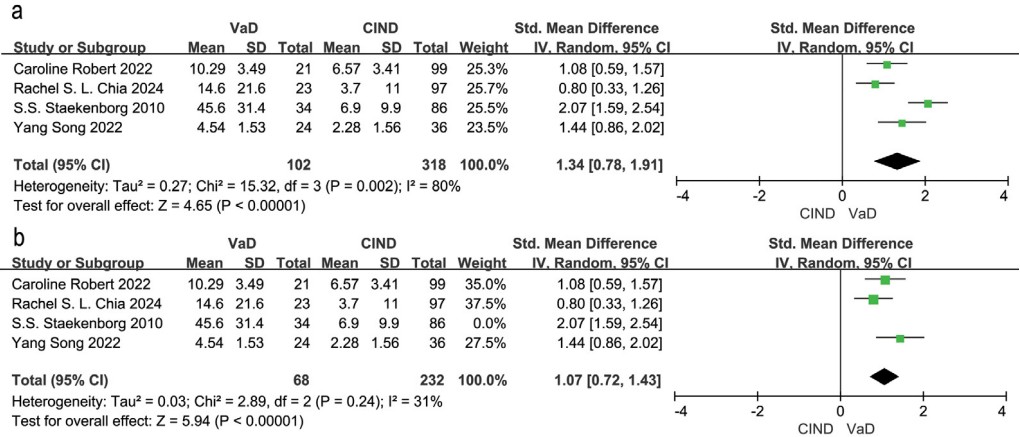

**Figure 3  Meta-analyses forest plot of WMH volume between patients with VaD and CIND.** Note. *Robert et al. (2022)*, *Chia et al. (2024)*, *Staekenborg et al. (2010)*, *Song et al. (2022)*.

Three studies (*Barber et al., 1999*; *Bokde et al., 2002*; *Varma et al., 2002*) could not be included in the meta-analysis due to insufficient data; however, their findings were generally consistent with the meta-analysis results. *Barber et al. (1999)* reported significantly higher WMH scores in the VaD group (WMH frontal score = 5) compared to the normal control group (WMH score = 0). *Bokde et al. (2002)* reported that WMH density was significantly higher in all four brain regions in VaD patients than in healthy controls ($P < 0.05$).

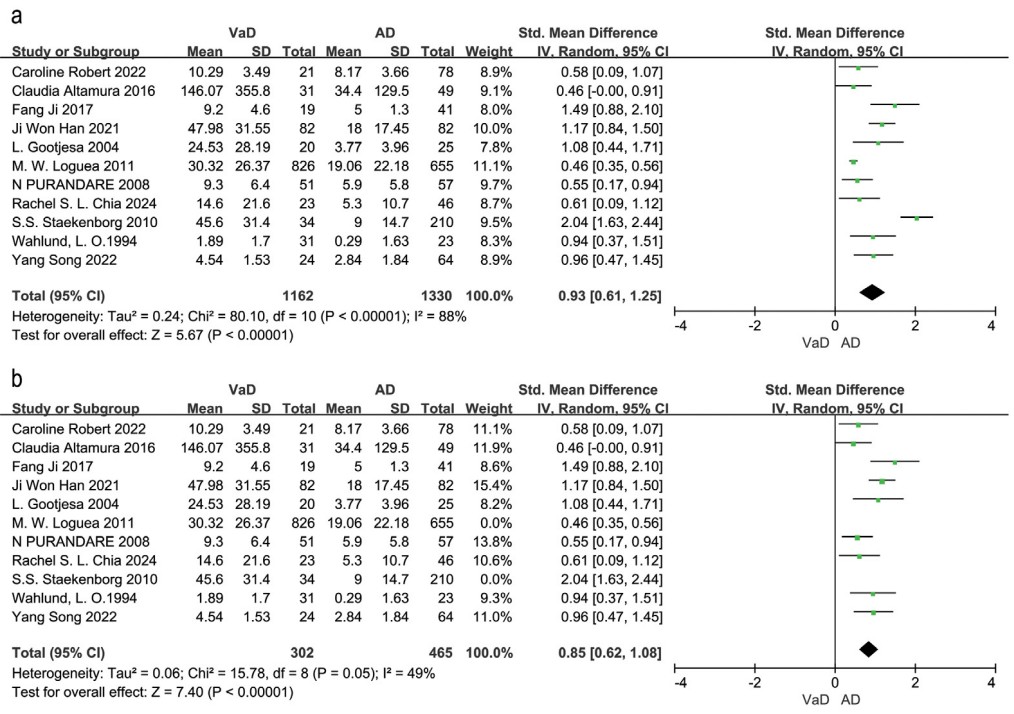

**Figure 4   Meta-analyses forest plot of WMH volume between patients with VaD and AD.** Note. *Robert et al. (2022), Altamura et al. (2016), Ji et al. (2017), Han et al. (2021), Gootjes et al. (2004), Logue et al. (2011), Purandare et al. (2008), Chia et al. (2024), Staekenborg et al. (2010), Wahlund et al. (1994), Song et al. (2022).*

Additionally, within each group, WMH density was significantly higher in the frontal and parietal lobes than in the temporal and occipital lobes ($P < 0.05$). *Varma et al. (2002)* reported that grade III deep white matter hyperintensities (DWMH) were specific to VaD when compared to normal controls and AD patients.

## Publication bias

Publication bias was assessed using funnel plots when more than 10 studies were included. The funnel plot comparing WMH volume between the VaD and AD groups is presented in Fig. 5, with four studies outside of the plot boundaries, suggesting potential publication bias. Egger's test further indicated possible publication bias or methodological heterogeneity across studies ($P = 0.043$, Fig. S1). Trim-and-fill analysis imputed six missing studies, which increased the pooled effect size from 0.644 (95% CI [0.559–0.728]) to 1.606 (95% CI [1.487–1.735]), suggesting that smaller negative studies may have been omitted from the initial analysis (Fig. S2).

## DISCUSSION

Recent meta-analyses and longitudinal studies have established that WMHs are associated with an increased risk of cognitive decline and dementia. A meta-analysis indicated that WMHs were linked to a 35% increased risk of progression from cognitively unimpaired

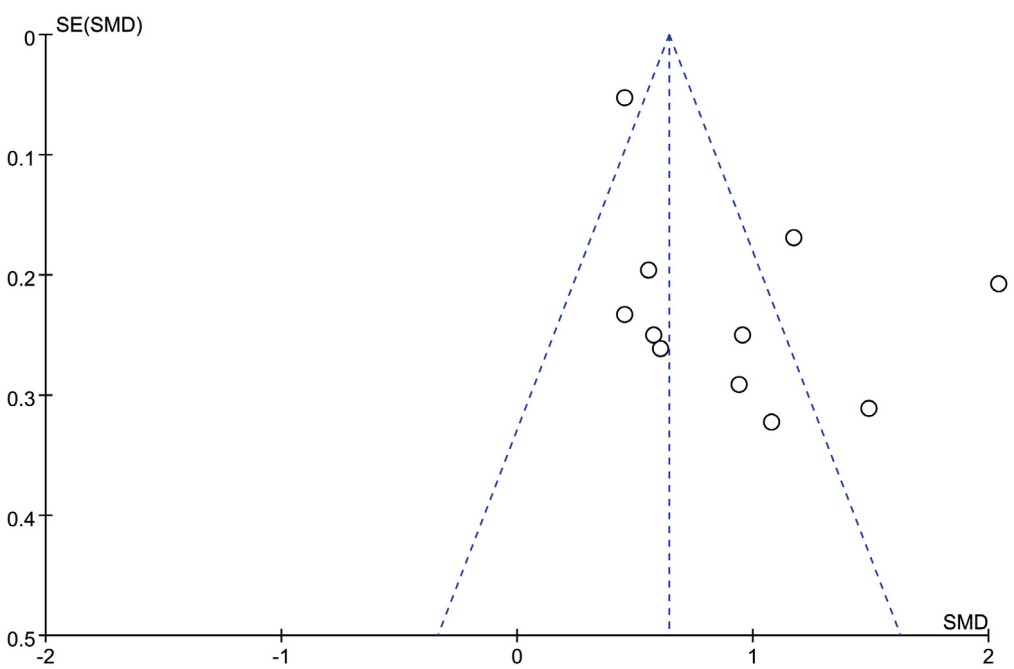

**Figure 5   Funnel plot of effect sizes for VaD group *vs.* AD group included in the meta-analysis.**

to mild cognitive impairment (MCI) and a 49% increased risk of progression to dementia (*Wahlund et al., 1994*). Additionally, Longitudinal evidence supports WMH progression as a prognostic biomarker in vascular cognitive impairment, with its expansion predicting accelerated cognitive decline, transitions to MCI and dementia, particularly in individuals harboring modifiable vascular risk factors such as hypertension (*Alber et al., 2019*). These findings highlight the importance of advanced imaging techniques in tracking WMH progression and its impact on cognitive decline. However, whether changes in WMH volume precede and independently predict VaD progression remains unclear. Our comprehensive meta-analysis confirms that higher WMH volume is associated with VaD, supporting that WMH-related vascular injury contributes to cognitive impairment.

One key objective of this study was to assess the association between WMH burden and VaD severity. Our findings indicate that patients in the VaD group exhibited significantly higher WMH volumes compared to patients in the AD, NCI, and CIND groups. This highlights the importance of WMH as a distinguishing feature of VaD, supporting its role in vascular cognitive impairment. Previous studies have shown that the extent of WMH correlates with global cognitive decline, executive dysfunction, and processing speed deficits, which are characteristic of VaD (*Cao et al., 2021*).

Additionally, WMHs are closely linked to carotid stiffness, a known risk factor for VaD. *Robert et al. (2022)* reported a significant association between carotid stiffness and WMH, suggesting that carotid stiffness may impair cerebral microcirculation, leading to WMH accumulation and ultimately contributing to VaD. As a marker of CSVD,
WMH may serve as a critical link between vascular pathology and cognitive impairment. Moreover, advancements in MRI segmentation techniques, such as the landmark-based fast brain region segmentation method described by *Bokde et al. (2002)* offer the potential for improved diagnostic accuracy in neurodegenerative and cerebrovascular diseases.

Beyond WMH volume, morphological characteristics also appear to play a role in VaD pathology. *Han et al. (2021)* reported that irregularly shaped WMHs are associated with cerebrovascular components of cognitive impairment. Compared to AD, VaD is more strongly linked to widespread deep and periventricular WMH lesions, which may reflect chronic hypoperfusion and small vessel disease. Similarly, *Purandare et al. (2008)* identified a significant negative correlation between DWMH severity and spontaneous cerebral emboli in VaD patients, suggesting that extensive ischemic contributes to VaD development. *Logue et al. (2011)* conducted a large-scale study involving 826 VaD patients and demonstrated that WMH severity correlates with cognitive decline, likely due to its impact on cerebral microcirculation. Their findings further reinforce the association between WMH burden, vascular pathology, and global brain atrophy in VaD patients.

To further clarify the role of WMH in VaD pathogenesis, we compared WMH volumes across different cognitive impairment groups. Our results revealed that: VaD patients had the highest WMH, supporting its association with small vessel disease. AD patients had lower WMH volumes than VaD but higher than NCI and CIND, suggesting a contribution of cerebrovascular pathology. CIND patients had intermediate WMH levels, indicating potential involvement before dementia onset. NCI patients had the lowest WMH burden, distinguishing pathological WMH accumulation from normal aging. These findings highlight WMH as a critical marker for differentiating cognitive impairment types, with irregular periventricular WMHs being particularly associated with cerebrovascular pathology.

To explore the pathophysiological mechanisms linking WMH and VaD, *Chen et al. (2016)* identified astrocyte dysfunction and BBB disruption as key contributors to post-stroke cognitive impairment and VaD. Their clinicopathological study in stroke patients with severe frontal WMH suggests that BBB dysfunction may exacerbate cognitive decline. Additionally, WMHs in the periventricular white matter, hippocampus, and associative cortical areas may disrupt neurotransmitter pathways, particularly cholinergic signaling, further contributing to cognitive impairment (*Chen et al., 2016*; *Kalheim et al., 2017*).

Despite the robust findings, publication bias and methodological heterogeneity must be considered when interpreting the pooled effect size for WMH volume differences between VaD and AD. Funnel plot asymmetry, corroborated by Egger's test, and the substantial upward adjustment of the effect size following trim-and-fill analysis suggest that smaller negative studies may be underrepresented in our meta-analysis. This potential selective reporting could lead to an overestimation of the true effect size, warranting cautious interpretation.

While WMHs serve as a valuable neuroimaging marker for VaD, they are not exclusive to this condition. Given the overlap with other forms of dementia, additional biomarkers are required to enhance diagnostic specificity. Future research should focus on: (1) Developing predictive models–Integrating WMH volume, shape, and distribution with

machine-learning algorithms could enhance early detection and risk assessment of VaD. (2) Longitudinal studies—Tracking WMH progression over time could clarify its causal role in VaD development and assess the impact of vascular risk factor management. (3) Multi-center studies with standardized MRI protocols—Variability in imaging techniques contributes to study heterogeneity. Standardized imaging and segmentation methods could improve comparability across studies. (4) Exploring genetic and systemic risk factors—Hypertension, diabetes, and chronic inflammation may accelerate WMH accumulation. Investigating these factors could inform targeted interventions. (5) Therapeutic implications—Given that WMHs may reflect cerebrovascular dysfunction, treatment strategies targeting vascular health (*e.g.*, antihypertensive therapy, and anti-inflammatory interventions) should be explored for their potential to slow cognitive decline.

This study has several limitations. First, most included studies were case-control and single-center, which may limit the generalizability of our findings. Second, while subgroup analyses could have provided further insights, we did not stratify by age, sex, or ethnicity due to limited sample sizes. Third, the potential for publication bias, as studies reporting no significant association between WMHs and VaD may be underrepresented. Lastly, heterogeneity in MRI acquisition protocols, segmentation methods, and VaD diagnostic criteria remains a challenge, underscoring the need for standardized research methodologies. The intended audience for this article includes medical professionals such as neurologists, geriatricians, and radiologists, as well as researchers in the fields of neurology, vascular medicine, and geriatric health. Additionally, it is relevant to public health policymakers and healthcare providers who are involved in the management and prevention of dementia and related neurological conditions.

## CONCLUSIONS

This study confirms that WMH volume is associated with VaD severity. Patients with VaD exhibit significantly greater WMH than those with AD, NCI, and CIND, reinforcing the role of cerebrovascular pathology in VaD. Beyond WMH volume, morphological characteristics such as shape and distribution may further differentiate VaD from other cognitive disorders. To enhance early diagnosis and intervention, future research should prioritize standardizing imaging assessments, integrating multi-modal biomarkers, and developing predictive models. These efforts could facilitate early detection, personalized risk assessment, and targeted therapeutic strategies for VaD. The study's findings on the relationship between WMH and VaD can also be of interest to medical students and trainees who are learning about neurodegenerative diseases.

**Abbreviations**

| | |
|---|---|
| **BBB** | Blood–brain barrier |
| **CI** | Confidence interval |
| **CSVD** | Cerebral small vessel disease |
| **MRI** | Magnetic resonance imaging |
| **VaD** | Vascular dementia |

| | |
|---|---|
| **WMH** | White matter hyperintensity |
| **NCI** | non-cognitively impaired |
| **CIND** | cognitively impaired no dementia |
| **AD** | Alzheimer's disease |
| **DWMH** | deep white matter hyperintensities |
| **MCI** | Mild cognitive impairment |

## ACKNOWLEDGEMENTS

We thank *Medjaden* Inc for their scientific editing of this manuscript.

### Funding
The authors received no funding for this work.

### Competing Interests
The authors declare there are no competing interests.

### Author Contributions
- Wei Luo performed the experiments, analyzed the data, prepared figures and/or tables, authored or reviewed drafts of the article, and approved the final draft.
- Zhiqiang Dai performed the experiments, analyzed the data, prepared figures and/or tables, authored or reviewed drafts of the article, and approved the final draft.
- Wenjing Wu performed the experiments, analyzed the data, prepared figures and/or tables, authored or reviewed drafts of the article, and approved the final draft.
- Haitao Li performed the experiments, analyzed the data, prepared figures and/or tables, authored or reviewed drafts of the article, and approved the final draft.
- Yang Zhang conceived and designed the experiments, performed the experiments, analyzed the data, prepared figures and/or tables, authored or reviewed drafts of the article, and approved the final draft.

### Data Availability
The data are available at Figshare: Yang Zhang (2025) Raw data and RevMan analysis. figshare. Dataset. https://figshare.com/articles/dataset/Raw_data_and_RevMan_analysis/28638329.

### Supplemental Information
Supplemental information for this article can be found online at http://dx.doi.org/10.7717/peerj.19460#supplemental-information.

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
