# Peer review of "White matter hyperintensities and the risk of vascular dementia: a systematic review and meta-analysis"

_PeerJ, doi:10.7717/peerj.19460_

## Round 0.1 · original submission · Major Revisions

Please revise this manuscript as per comments of the two peer reviewers.

Reviewer 1 ·

Basic reporting

✅ The manuscript is well-written and generally follows professional scientific language. However, some sentences are unnecessarily complex, which may reduce readability. Certain grammatical inconsistencies (e.g., "Researche" instead of "Research" in the Introduction) should be corrected. I recommend a thorough proofread to enhance clarity.

✅ The authors provide a solid introduction to white matter hyperintensities (WMHs) and their implications in vascular dementia (VaD). However, the literature review lacks some recent meta-analyses and longitudinal studies on WMH progression in dementia. Including more references on multi-modal MRI studies and advanced quantitative imaging biomarkers would strengthen the discussion.

✅ The structure adheres to standard formatting, with well-defined sections. Figures are appropriately labeled. Also, the authors acknowledge publication bias in Figure 5 (funnel plot), but no corrective statistical measures (such as Egger's test) are applied. This should be addressed.

✅ The manuscript states that "all data generated or analyzed during this study are included in this published article," but there is no direct link or repository reference. If possible, raw data and RevMan analysis files should be made available for transparency.

Experimental design

✅ The study addresses a relevant and clinically significant question within the scope of vascular dementia and small vessel disease. However, it does not provide novel insights beyond confirming existing associations between WMH burden and VaD risk. The authors should clarify how their meta-analysis adds to the current body of knowledge.

✅ The methodology is clearly described, adhering to PRISMA guidelines. However, some concerns include:
1) The search strategy is limited to three databases (MEDLINE, EMBASE, and Cochrane), excluding sources like Scopus and Web of Science, which could introduce selection bias.
2) The inclusion/exclusion criteria are reasonable, but the rationale for excluding non-English studies is not provided. This may contribute to a language bias.
3) The diagnostic criteria for VaD vary across included studies (e.g., NINDS-AIREN vs. DSM-IV). Were efforts made to ensure compatibility between different diagnostic methods?

✅ The use of standardized mean difference (SMD) and 95% confidence intervals (CI) is appropriate, but the heterogeneity (I² = 72–88%) is high. The authors attempt sensitivity analysis but should further discuss possible sources of heterogeneity (e.g., differences in MRI protocols, segmentation methods).

✅ The funnel plot suggests publication bias, but this is not rigorously tested using Egger’s regression or trim-and-fill analysis. These should be included to assess the robustness of the conclusions.

✅ The methodology is well-documented but lacks details about how WMH volumes were measured across studies. Were all studies using fully automated segmentation? If not, how was bias controlled? These aspects should be clarified for reproducibility.

Validity of the findings

✅ The results support the conclusion that WMH burden is associated with VaD, but given the small sample size (only seven studies with 4061 patients), caution should be taken when generalizing the findings. The authors acknowledge potential bias but do not provide enough discussion on whether certain subgroups (e.g., age, sex, ethnicity) might influence WMH-VaD associations.

✅ The discussion does not fully address causality—is WMH volume merely a biomarker, or does it contribute mechanistically to VaD progression?

✅ The authors mention that WMH shape and distribution are also relevant, but these are not analyzed systematically in this meta-analysis. If these factors are important, should future studies include topographic WMH analysis?

✅ No distinction between periventricular vs. deep WMHs is made, despite evidence suggesting different pathophysiological mechanisms. A subgroup analysis on this aspect would be beneficial.

✅ The conclusion is somewhat overstated. While the results confirm a correlation, they do not establish a direct causal link between WMH and VaD. The authors should moderate their claims and acknowledge that other vascular risk factors (hypertension, diabetes, stroke history) might mediate this relationship.

Additional comments

✅ The study’s findings have potential clinical implications for identifying high-risk patients. However, the manuscript does not provide a clear practical application of WMH volume as a predictive marker. Could the authors discuss possible clinical cut-off values for risk stratification in dementia prevention?

✅ The authors briefly mention future studies but do not specify concrete next steps. Suggestions could include:
1) Multi-center studies using harmonized MRI protocols.
2) Advanced machine-learning techniques for WMH segmentation and prediction models.
3) Inclusion of longitudinal data to track WMH progression over time.

✅ Table 1 (Characteristics of Included Studies) contains inconsistent formatting, making it harder to interpret the data. Standardizing column widths and text alignment would improve readability.

✅ Sentence repetition (e.g., discussion of WMH shape appears in multiple sections). The manuscript would benefit from a more concise structure.

✅ The study is well-conceived and methodologically sound, but several major concerns need to be addressed:
1) Improve the discussion on heterogeneity and confounders.
2) Perform additional bias correction analyses (Egger’s test, trim-and-fill).
3) Clarify how WMH measurement methods were standardized.
4) Moderate claims regarding causality.
5) Provide clearer clinical applications and next research steps.

✅ Once these issues are addressed, the manuscript will contribute meaningfully to the field of cerebrovascular aging and dementia research.

Reviewer 2 ·

Basic reporting

The manuscript entitled "White matter hyperintensities and the risk of vascular dementia: A systematic review and meta-analysis” performed a metaanalysis of studies on white matter hyperintensities and vascular dementia.

The Introduction needs to include more current and detailed information on vascular dementia. The Prisma flowchart needs a recheck of the numbers as they do not add up correctly, and needs to give the reasons for exclusion, while the discussion needs to focus more on discussing its findings, and to clearly answer all of its objectives.

Experimental design

No comment

Validity of the findings

Please check the numbers in Prisma flowchart as they do not add up correcty.

Additional comments

Title: Appropriate
Abstract:
Method: There is discrepancy in the number of studies between abstract (7) and full text (15).
Results: Please include the statistical test results.

Introduction
For the most part in the introduction, references are outdated and provided as monoreference, which does reflect the currentness of the subject matter. Not much information is given on vascular dementia which is the crux of the meta analysis (see Bir 2022 for a review)
Line 63 please provide reference
Line 64 spelling Researche
Line 77 However, other studies have found no significant relationship… please provide reference

Methods
Line 90 the reference for Cochrane handbook should be current
Line 102 typo WHM
Line 134 include the date until when each database was searched

Results

Line 145 Meta-analysis and heterogeneity analysis. Please use superscipt for '2' in I2

Line 182 ..grade III DWMH . Define DWMH and put in abbreviation list



Discussion

The line of discussion does not really follow the objectives of the study, i.e. to evaluate the association between WMH volume and VaD severity, to clarify the
significance of WMH for VaD pathogenesis, to compare the differences in WMH volumes among VaD patients, AD patients, non-cognitively impaired (NCI) patients and cognitively impaired no dementia (CIND) patients. While the results of the first objective were extensively discussed, there was hardly any discussion of the other findings, as well as the demographics of the patients.

Line 192-197 The first three sentences in Discussion are better placed in Introduction as they are not discussion of the current study.

Line 245 Caroline Robert reported….
249 A.L.W. Bokde….
Please correct the in text citation format

Line 255 Define SCE


Supplemental materials

PRISMA flowchart
Please check the numbers as they do not sum up correctly, and give reasons for the excluded studies.

---

## Round 0.2 · accepted · Accept

Thank you for your revised manuscript, which has been accepted.

Reviewer 1 ·

Basic reporting

1. Unambiguous, professional English used throughout.
2. Sufficient literature references and field background/context provided.
3. Professional article structure, figures, and tables.

Experimental design

The authors addressed and improved all necessary/ required information. I have no further comments

Validity of the findings

No comment
The findings are valid

Reviewer 2 ·

Basic reporting

The authors have made appropriate corrections to the manuscript, and had significantly improved its overall presentation in terms of language and scientific writing. I have no further comments.

Experimental design

No comment

Validity of the findings

No comment